# Assessing Carbon Emissions from Animal Husbandry in China: Trends, Regional Variations and Mitigation Strategies

Cheng Peng [1], Xiaona Wang [1], Xin Xiong [2,*] and Yaxing Wang [3]

[1] School of Economics and Management, Southwest University of Science and Technology, Mianyang 621010, China; pengcheng@swust.edu.cn (C.P.); xnwswust@163.com (X.W.)
[2] Faculty of Geosciences and Environmental Engineering, Southwest Jiaotong University, Chengdu 611756, China
[3] School of Life Science and Engineering, Southwest University of Science and Technology, Mianyang 621010, China; wangyaxing@mails.swust.edu.cn
* Correspondence: xiongxin@swjtu.edu.cn

**Abstract:** The intensification of land use and the accelerated integration of three industries (agricultural production, agricultural product processing industry, and agricultural product market service industry) in China have resulted in the continuous expansion of animal husbandry and its industrial chain. This phenomenon has led to a rise in greenhouse gas (GHG) emissions from livestock farming, intensifying climate change and placing strain on worldwide environmental conservation efforts. Life cycle assessment (LCA) was utilized to evaluate carbon emissions from China's animal husbandry sector from 2012 to 2021. Additionally, the logarithmic mean divisia index (LMDI) decomposition method was employed to examine and elucidate the influential impacts of five factors on carbon emissions. These factors included the efficiency of animal husbandry production, the structure of the agricultural industry, per capita agricultural production income, urbanization, and the total population. The results reveal the following: (1) From 2012 to 2021, China's animal husbandry sector witnessed a marginal increase in cumulative carbon emissions from 287.74 million tons to 294.73 million tons, with an annual growth rate of 0.42%. (2) Emission contributions were categorized as follows: the production stage (149.61 million tons), the transportation stage (145.07 million tons), and the processing stage (0.05 million tons). (3) The primary factor contributing to the rise in carbon emissions from animal husbandry from 2012 to 2021 was the per capita agricultural production income factor ($A_3$), alongside a notable impact from the total population factor ($A_5$).

**Keywords:** GHG emissions; LMDI method; LCA method; driving factors of carbon emissions; livestock production stages

## 1. Introduction

The effects of climate warming on the survival and development of human beings have been widely recognized [1,2]. An urgent global concern for all countries is the effective reduction in greenhouse gas (GHG) emissions [3]. Livestock production significantly contributes to global emissions of $CH_4$, $N_2O$, $CO_2$, and $NH_4$ [4]. China is a prominent agricultural nation, boasting the most extensive animal husbandry resources worldwide. In recent years, the animal husbandry sector in China has undergone significant growth and has progressively established itself as a crucial economic pillar industry in the country. The global production of various livestock and poultry products has consistently maintained its leading position for many years [5]. By 2021, projections indicate that China's animal husbandry sector will achieve an output value of CNY 3991.08 billion, representing 27.15% of the nation's total agricultural production. Simultaneously, carbon emissions originating from animal husbandry have become a significant factor in China's total carbon emissions [6]. The intensification of land use and the accelerated integration of three industries (agricultural production, agricultural product processing industry, and agricultural product

market service industry) have resulted in animal husbandry being expanded continuously, together with its downstream industrial chain. With regard to carbon emissions, China established two goals in 2020: "achieving carbon neutrality by 2030 and peaking emissions by 2060." This analysis aims to provide China's animal husbandry industry a scientific foundation for achieving green and low-carbon transformation at the industrial level to help achieve carbon neutrality and the lowest emissions possible, and to support the sustainable growth of the nation's economy and society [7,8]. Furthermore, a precise assessment of carbon emissions resulting from animal husbandry, along with an identification of factors that influence these emissions and measures to mitigate them, can serve as a foundation for decision-making when China's animal husbandry sector shifts to low-carbon, ecologically friendly practices. Establishing a sustainable agriculture system and advancing the dual-carbon goal's realization from a consumer standpoint are very important for China.

## 2. Literature Review

### 2.1. Quantification and Assessment of Livestock Carbon Emissions

It is acknowledged that livestock contributes significantly to carbon emissions worldwide [5]. Numerous national and international studies have evaluated its related carbon emissions quantitatively [9–11]. Hao et al. quantified carbon emissions associated with intestinal fermentation and fecal management in animal husbandry practices across different provinces in China. Their research also encompassed an examination of the temporal and spatial trends of carbon emissions originating from animal husbandry [10]. Yao et al. evaluated the quantity of carbon emissions linked to livestock and poultry, covering the complete process from providing feed to managing manure [12]. The utilization of LCA has been widespread in the realms of animal husbandry and carbon emission accounting, particularly in relation to certain categories of livestock and poultry products. This method has shown significant applicability in research practices [13]. The environmental effects of feed production and transportation, broiler house operations, and broiler production manure management were examined by Ogino et al. using life cycle analysis [14]. Using a life cycle assessment methodology, Rosen et al. assessed the carbon emissions of the entire cattle business chain, accounting for feed cultivation, meat processing, and transportation [15]. Livestock and poultry products are the main source of greenhouse gas emissions, according to research on carbon emissions from livestock. This encompasses the supervision of feed grain, the intestinal tract and feces of livestock and poultry, and the energy consumption involved in feeding [16–18].

### 2.2. Analysis the Influencing Factors of Carbon Emissions

For the examination of factors that influence carbon emissions, current research has predominantly employed structural decomposition analysis (SDA) and the index decomposition approach (IDA) [19,20]. LMDI stands out for its ability to generate a residual of zero following decomposition, resulting in a different outcome, while being relatively easy to use. Consequently, it has been widely applied in various fields in recent years [21–23]. Wang et al. employed SDA to examine the influence of economic factors, population growth, and carbon emission intensity on carbon emissions [24]. Their study's findings indicate that although carbon intensity is the primary factor regulating carbon emissions, population growth and the economy are the primary factors causing emissions to rise. Yang et al. conducted an analysis of the impact of population size, economic development, industrial structure, energy intensity, and energy structure on carbon emissions in Hunan Province using an LMDI model [25]. According to the findings, Hunan Province's economic expansion has the biggest influence on rising carbon emissions, while energy consumption intensity is the biggest deterrent. Previous research has demonstrated that a variety of factors, including population increase, economic structure, economic growth, and per capita income, have an impact on carbon emissions. He et al. applied the LMDI decomposition method to analyze the influence of efficiency, structure, economy, and population size on carbon emissions within this industry [26]. The results show that while population

size, farmer income level, and industrial structure sectors all significantly promote carbon emissions from animal husbandry, urbanization and the degree of agricultural mechanization have a notable inhibiting effect on these emissions. Bretscher et al. emphasized the substantial significance of restructuring agricultural practices to mitigate emissions within the animal husbandry sector [27]. Dai et al. employed the LMDI decomposition method for analyzing the total carbon emissions from animal husbandry in China [28]. The findings show that while changes in the structure of the national industry, agricultural practices, and technology have all served as deterrents to carbon emissions, population growth and economic expansion have had a positive impact on carbon emissions.

*2.3. Summary*

In summary, the separation of production and consumption brought about by increased urbanization has resulted in the expansion of the livestock industry's downstream industrial chain. Nevertheless, the majority of recent research studies on estimating carbon emissions from animal husbandry have focused on the breeding stage of livestock and poultry, while paying little attention to downstream processing, particularly transportation [12,14]. The total carbon emissions from the livestock industry could be significantly underestimated if transportation is excluded. Furthermore, the economic benefits of livestock production outweigh those of the plantation business when it comes to factors that influence carbon emissions from the livestock industry. Livestock producers may alter the industrial structure of agriculture and increase the volume of livestock production in an attempt to increase production income due to profit-seeking motives. So, how do carbon emissions from livestock relate to the structure of the agricultural sector and the per capita income from agricultural production? How will the fast-rising urbanization trend in China impact the livestock industry's carbon emissions as more people move from rural to urban areas? Consequently, the production, processing, and transportation stages of animal products were designated as the system boundaries, and a process-based life cycle assessment (PLCA) model from the "cradle" to the "gate" was built in this study. The regional and temporal aspects of their changes were uncovered by analyzing the carbon emissions of several key livestock and poultry species in China from 2012 to 2021 over their whole life cycle, from breeding to consumption. Based on this, the LMDI method was used to explore the effects of factors such as total population, urbanization, per capita income from agricultural production, livestock production efficiency, and agricultural industry structure in the livestock industry. The use of this method helps to achieve carbon neutrality while offering a scientific foundation for China's livestock industry's transition to a low-carbon and greener state at the industry level.

**3. Methodology**

*3.1. Accounting of Livestock Production Carbon Emissions*

LCA was utilized to measure carbon emissions linked to animal husbandry. The system boundary comprised three stages: production, processing, and transportation. The main emphasis was placed on pigs, cattle, sheep, poultry, eggs, milk, and other significant livestock products.

3.1.1. Production Stage

Carbon emissions during the production phase of animal husbandry primarily stem from activities such as planting, transporting, and processing raw materials for animal husbandry, as well as livestock and poultry manure management and energy consumption according to gastrointestinal fermentation.

(1)    Feed grain planting

Livestock and poultry feeds are predominantly classified into two main categories: roughage and concentrated feed. Roughage, comprising by-products like corn stalks, rice straw, and sweet potato vines, is produced via a single processing stage and results in minimal carbon emissions; therefore, we excluded it from consideration [12]. Concentrated

feed primarily consists of corn, soybeans, and wheat. Based on the carbon emissions that result from chemical fertilizers, pesticides, and agricultural films, the energy required to produce concentrated feed grows. The specific calculation equation is as follows:

$$TC_{fp} = \sum_{u=1}^{n} Q_u \cdot s_u \cdot q_j \cdot ef_{j1} + t_j \cdot m_j \cdot B_j \cdot ef_h \tag{1}$$

where $TC_{fp}$ represents carbon emissions produced from feeding grain crops during planting. u represents the types of livestock products, including beef, mutton, pork, poultry, eggs, and milk. $Q_u$ represents the annual output of class u with livestock products (t). $s_u$ represents the grain consumption coefficient of class u livestock and poultry products for class j grain (kg/kg), and $q_j$ represents the proportion of class j grain in the livestock and poultry (%) feed formula, primarily including corn, wheat, and soybean (Table 1). $ef_{j1}$ represents the $CO_2$ equivalent emission coefficient of class j feed grain during planting (t/t) (Table 2). GHG emissions generated from soybean planting are not included in the calculation, as bean cake is a by-product of soybeans' primary processing [12]. $t_j$ represents the amount of a chemical fertilizer used per unit area of class j grain (kg/hm$^2$), $m_j$ represents the sown area of class j grain (hm$^2$), $B_j$ represents the proportion of class j grain (%), and $ef_h$ represents the $CO_2$ emission coefficient of the fertilizer.

**Table 1.** Feed proportion analysis of livestock and poultry products.

| Livestock Products | Ratio | | | Reference |
|---|---|---|---|---|
| | **Corn** | **Soybean** | **Wheat** | |
| Pig | 56.6% | 10.2% | - | |
| Cattle | 37% | 26% | - | |
| Sheep | 62.61% | 12.89% | - | |
| Broiler | 57% | 17% | 5% | [12] |
| Laying hen | 63.28% | 13.98% | - | |
| Dairy cattle | 46.79% | 28.56% | - | |

**Table 2.** GHG emission factors at each stage.

| System | Symbol | Name | Number | Unit | References |
|---|---|---|---|---|---|
| Feed grain cultivation | $ef_{j1}$ | $CO_2$ emission coefficient of maize | 1.50 | t/t | [12] |
| | | $CO_2$ emission coefficient of wheat | 1.22 | t/t | |
| | $ef_h$ | $CO_2$ emission coefficient of fertilizer | 3.283 | t/t | |
| Feed grain processing and transportation | $ef_{j2}$ | $CO_2$ emission coefficient of maize | 0.0102 | t/t | [12] |
| | | $CO_2$ emission coefficient of soybean | 0.1013 | t/t | |
| | | $CO_2$ emission coefficient of wheat | 0.0319 | t/t | |
| Livestock and poultry raising | $P_e$ | Unit price of electricity for raising | 0.4275 | CNY/kWh | [12] |
| | $ef_e$ | $CO_2$ emission coefficient of electric consumption | 0.9734 | t/MWh | |
| | $P_c$ | Unit price of coal for raising | 800 | CNY/t | |
| | $ef_c$ | $CO_2$ emission coefficient of coal consumption | 1.98 | t/t | |

**Table 2.** *Cont.*

| System | Symbol | Name | Number | Unit | References |
|---|---|---|---|---|---|
| Livestock and poultry product processing | $MJ_u$ | Energy consumption coefficient of pork slaughtering and processing | 3.76 | MJ/kg | [29] |
| | | Energy consumption coefficient of beef slaughtering and processing | 4.37 | | |
| | | Energy consumption coefficient of mutton slaughtering and processing | 10.4 | | |
| | | Energy consumption coefficient of poultry slaughtering and processing | 2.59 | | |
| | | Energy consumption coefficient of milk processing | 1.12 | | |
| | | Energy consumption coefficient of egg processing | 8.16 | | |
| Others | $e_n$ | One-degree calorific value | 3.60 | MJ | [12] |
| | $e_{tpf}$ | Carbon coefficient of $CO_2$ | 0.2728 | - | |
| | $GWP_{CH_4}$ | Global warming potential of $CH_4$ | 21 | | |
| | $GWP_{N_2O}$ | Global warming potential of $N_2O$ | 310 | | |
| | $e_t$ | Carbon emission coefficient of standard coal | 0.68 | | [30] |

(2) Feed grain transportation and processing

Corn, soybean, wheat, and other food crops require a series of processing steps including cleaning, drying, crushing, and mixing with other ingredients, as well as various transportation steps, before being transformed into feed for livestock and poultry. This phase primarily concentrates on GHG emissions generated from energy consumption during transportation and processing operations [12]. The equation for calculating $CO_2$ emissions associated with the transportation and processing of a feed grain is as follows:

$$TC_{tp} = \sum_{u=1}^{n} Q_u \cdot s_u \cdot q_i \cdot ef_{j2} \tag{2}$$

where $TC_{tp}$ represents the $CO_2$ emission of a feed grain during the transportation and processing stages; $s_u$ represents the grain consumption coefficient of class u-type livestock and poultry products of the unit (kg/kg); $q_j$ represents the proportion of class j grain, primarily including corn, wheat, and soybean, in the livestock and poultry feed formula (%) shown in Table 1; and $ef_{j2}$ represents the equivalent emission coefficient of class j grain during the processing and transportation stages.

(3) $CH_4$ emissions from gastrointestinal fermentation

Gastrointestinal fermentation in livestock and poultry leads to the generation of GHGs, such as $CH_4$. Rumen fermentation in ruminant livestock, such as cattle and sheep, constitutes the primary source of $CH_4$ emissions. It accounts for more than 80% of the total gas emitted from the intestines of all livestock and poultry. Non-ruminant livestock, such as horses, mules, and donkeys, and monogastric animals, like pigs, produce negligible quantities of $CH_4$ as a by-product of gastrointestinal fermentation. Furthermore, the minimal $CH_4$ emissions produced through the gastrointestinal fermentation process in poultry are not considered in the scope of this study [12]. The equations for calculating $CH_4$ emissions resulting from the gastrointestinal fermentation processes in both livestock and poultry are as follows:

$$TC_{sf} = \sum_{i=1}^{n} APP_i \cdot ef_{i1} \cdot GWP_{CH_4} \tag{3}$$

$$APP_i = \begin{cases} Herds_{end}, & if : Days_{live} \geq 1a \\ Days_{live} \cdot \left(\frac{NAPA}{365}\right), & if : Days_{live} \leq 1a \end{cases} \tag{4}$$

where $TC_{sf}$ represents the carbon emission produced by the gastrointestinal fermentation of a livestock, and i represents the type of livestock. $APP_i$ represents the class i livestock's annual average number (head/100). $ef_{i1}$ represents the emission coefficient of $CO_2$ produced by the gastrointestinal fermentation of class i livestock (Table 3). $GWP_{CH_4}$ represents the global warming potential of $CH_4$ (Table 2). $Herds_{end}$ represents the year-end inventory (head/100). $Days_{live}$ represents the livestock feeding cycle (das), and NAPA denotes the rate of livestock slaughter (head/100) in one year.

**Table 3.** $CH_4$ emission coefficient of gastrointestinal and fecal management.

| Species | Emission Coefficient of $CH_4$ | | Reference |
| --- | --- | --- | --- |
| | Gastrointestinal Fermentation ($ef_{i1}$) | Fecal Management ($ef_{i2}$) | |
| Pig | 1 | 3.50 | |
| Sheep | 5.00 | 0.16 | [12] |
| Cattle | 68 | 16 | |
| Poultry | 0 | 0.02 | |

(4)    Carbon emissions from manure management system

Due to the different GHGs produced during fecal degradation in anaerobic and aerobic environments, this study was divided into two sections to evaluate GHG emissions from the fecal management system.

The $CO_2$ emissions in an anaerobic environment can be determined as follows:

$$TC_{fc} = \sum_{i=1}^{n} APP_i \cdot ef_{i2} \cdot GWP_{CH_4} \tag{5}$$

where $TC_{fc}$ represents the carbon emission generated by the manure management system in an anaerobic environment, and $ef_{i2}$ represents the $CH_4$ emission coefficient of class i livestock in the manure management system (Table 3).

The $CO_2$ emission in an aerobic environment can be expressed as follows:

$$TC_{fn} = \sum_{i=1}^{n} APP_i \cdot ef_{i3} \cdot GWP_{N_2O} \tag{6}$$

where $TC_{fn}$ represents the carbon emission generated by the manure management system in an aerobic environment, $GWP_{N_2O}$ represents the global warming potential of $N_2O$ (Table 2), and $ef_{i3}$ represents the $N_2O$ emission coefficient of class i livestock in the system of manure management (Table 3).

(5)    Energy consumption in the feeding chain

Raising livestock and poultry requires substantial quantities of electricity and coal for various operations, including lighting, ventilation, and heating within barn facilities [31,32]. The equation for calculating carbon emissions caused by energy consumption is as follows:

$$TC_{rc} = \sum_{i=1}^{n} APP_i \cdot \frac{C_{ie}}{P_e} \cdot ef_e + \sum_{i=1}^{n} APP_i \cdot \frac{C_{ic}}{P_c} \cdot ef_c \tag{7}$$

where $TC_{rc}$ is the carbon emissions caused by the energy used to raise cattle and poultry. $C_{ie}$ and $C_{ic}$ denote the unit consumption expenditure of electricity and coal by class i livestock during its feeding cycle (CNY/head), respectively. $P_e$ and $P_c$ represent the $CO_2$ emission coefficient of electric energy consumption and coal consumption, respectively (Table 2). $ef_c$ represents the $CO_2$ emission coefficient (Table 2).

The carbon emissions produced during the production stage are quantified as follows:

$$TC_P = \left( TC_{fp} + TC_{tp} + TC_{sf} + TC_{fc} + TC_{rc} \right) \cdot e_{tpf} \tag{8}$$

where $TC_P$ represents the carbon emissions of the livestock production stage, and $e_{tpf}$ represents the conversion of an equivalent into a standard carbon coefficient, as shown in Table 2.

### 3.1.2. Processing Stage

Livestock and poultry are subjected to live slaughtering and processing procedures before being deemed suitable for market consumption. This process requires significant energy use, resulting in substantial carbon emissions. The equation for calculating the emissions from this livestock and poultry product process is as follows:

$$TC_A = \sum_{u=1}^{n} Q_u \cdot \frac{MJ_u}{e_n} \cdot ef_e \cdot e_{tpf} \tag{9}$$

where $TC_A$ represents the carbon emissions from the processing stage of livestock and poultry products; $Q_u$ represents the annual output of class u livestock products; u represents the class types of livestock products, including pork, beef, milk, mutton, poultry meat, and eggs; $MJ_u$ represents the energy consumption coefficient of the unit livestock products during the processing stage; $e_n$ represents the calorific value generated per degree of electricity consumption; and $ef_e$ represents the emission coefficient of electric energy consumption (Table 2).

### 3.1.3. Transportation Stage

The carbon emissions associated with livestock products during transportation and storage primarily encompass the energy consumption related to transportation vehicles and the use of storage agents for cold storage and heat preservation during transportation. The calculation equation is as follows:

$$TC_T = \frac{p \cdot F}{V_T} \cdot L_T \cdot e_t \tag{10}$$

where $TC_T$ represents the livestock product-generated carbon emissions during the transportation stage, and p represents the proportion of the livestock products' logistics cost relative to the food price (%). F represents the consumption expenditure of various animal products (ten thousand yuan), and $L_T$ denotes the transportation industry's whole energy consumption (in ten thousand tons of standard coal value). $V_T$ denotes the total output value of the energy transportation industry (ten thousand yuan). $e_t$ represents the carbon emission coefficient of standard coal (Table 2).

Therefore, the total carbon emission from animal husbandry can be expressed as follows:

$$TC_{TOTAL} = TC_P + TC_A + TC_T \tag{11}$$

where $TC_P$, $TC_A$, and $TC_T$ represent the carbon emissions of the livestock system during the production stage, the processing stage, and the transportation stage, respectively.

### 3.2. Influence of Different Factors on Carbon Emission of Livestock System

Building on the results of the carbon emission computation that was previously conducted, this study used an LMDI decomposition model to examine carbon emissions from animal husbandry. This study further investigated the influence of five factors, namely the efficiency of animal husbandry production ($A_1$), the structure of agricultural production ($A_2$), per capita income from agricultural production ($A_3$), urbanization ($A_4$), and total

population ($A_5$), on the carbon emissions from animal husbandry [33]. The calculation method is as follows:

$$D = TC^T/TC^0 = D_{A_1} \cdot D_{A_2} \cdot D_{A_3} \cdot D_{A_4} \cdot D_{A_5} \qquad (12)$$

$$\Delta TC = TC^T - TC^0 = \Delta A_1 + \Delta A_2 + \Delta A_3 + \Delta A_4 + \Delta A_5 \qquad (13)$$

where D and $\Delta TC$ represent the change in growth rate and growth amount of carbon emissions after the T period, respectively. $A_1$ represents the ratio of livestock carbon emissions to livestock GDP, $A_2$ represents the ratio of livestock GDP to agricultural GDP, $A_3$ represents the ratio of agricultural GDP to agricultural population, $A_4$ represents the ratio of agricultural population to national population, and $A_5$ represents the total population of the country.

*3.3. Data Source and Processing*

The data utilized in this study encompassed various parameters, such as the output values of animal husbandry, agriculture, forestry, and fishery; the sown areas of wheat, soybean, and corn; the annual output of livestock and poultry; the rate of livestock and poultry slaughter, and the year-end stock data from 2012 to 2021 across 30 provinces (cities and districts) in China. These data were extracted from the *Rural Statistical Yearbook of China*, spanning from 2013 to 2022. The transportation GDP statistics were derived from the *Statistical Yearbook of China*, spanning from 2013 to 2022. Meanwhile, data on electricity consumption, coal consumption per unit of livestock and poultry, grain consumption coefficients per unit of livestock products, and fertilizer application per unit of wheat, corn, and soybeans were obtained from the National Collection of Cost–Benefit Data of Agricultural Products (2013–2022). This study employed the linear interpolation method and the average growth rate method to address the absence of certain data regarding residents' food consumption.

**4. Results**

*4.1. Temporal Variation in GHG Emissions*

Figure 1 depicts the evolution and primary stages of carbon emissions from animal husbandry, highlighting a 2.43% increase in total emissions from 287.74 million tons in 2012 to 294.73 million tons in 2021. This increment reflects an average annual growth rate of 0.42%. In 2021, the carbon emission analysis revealed that the production stage was the largest contributor with 149.61 million tons, followed by the transportation stage with 145.07 million tons, and the processing stage with 0.049 million tons. The transportation stage was identified as the predominant source of carbon emissions from animal husbandry, accounting for 45.78% to 55.21% of the total carbon emissions. The production stage also makes a significant contribution, accounting for approximately 44.78% to 54.20% of the total carbon emissions. The contribution ratios of the processing stage to total carbon emissions are relatively small and show minimal variation, with an average annual ratio of 0.02%.

*4.2. Spatial Heterogeneity of Carbon Emissions*

The spatial distribution of livestock carbon emissions is shown in Figure 2. The regional ranking of average annual total carbon emissions is as follows: the western region (109.58 million tons) has the highest emissions, followed by the central region (69.81 million tons), the eastern region (60.94 million tons), and the northeast region (42.01 million tons). In the western region, carbon emissions are the highest due to animal husbandry, accounting for approximately 38.81% of the national total, because of the abundant resources in the western provinces' animal husbandry sector, including in Inner Mongolia, Sichuan, and Qinghai, leading to a significant overall scale in terms of carbon emissions in these areas due to animal husbandry. These areas are primarily dominated by small-scale retail farming operated by families, with farmers and herdsmen who have limited education and a weak awareness of environmental protection [10]. Consequently, the diminished production

efficiency in animal husbandry has intensified carbon emissions in the western region. The importance of the eastern and central regions' role in carbon emissions from animal husbandry is significant and should not be underestimated. In 2021, the carbon emissions from animal husbandry in the western region comprised 39.10% of the national total, with the central region and the eastern region contributing 23.62% and 19.37% to the national total, respectively. From 2012 to 2021, there was minimum fluctuation in the proportion of carbon emissions in the northeast region of China, with an average annual carbon emission accounting for approximately 14.84%. In 2021, Heilongjiang Province, Sichuan Province, Liaoning Province, Shandong Province, and Yunnan Province were the top five regions in terms of carbon emissions (Figure 3).

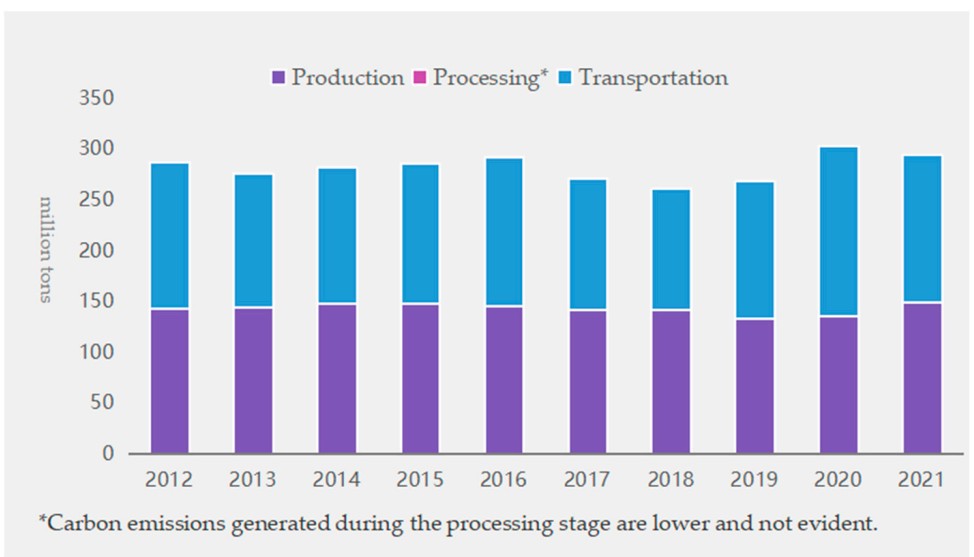

**Figure 1.** The decade carbon emission trends from the livestock industry in China (2012–2021).

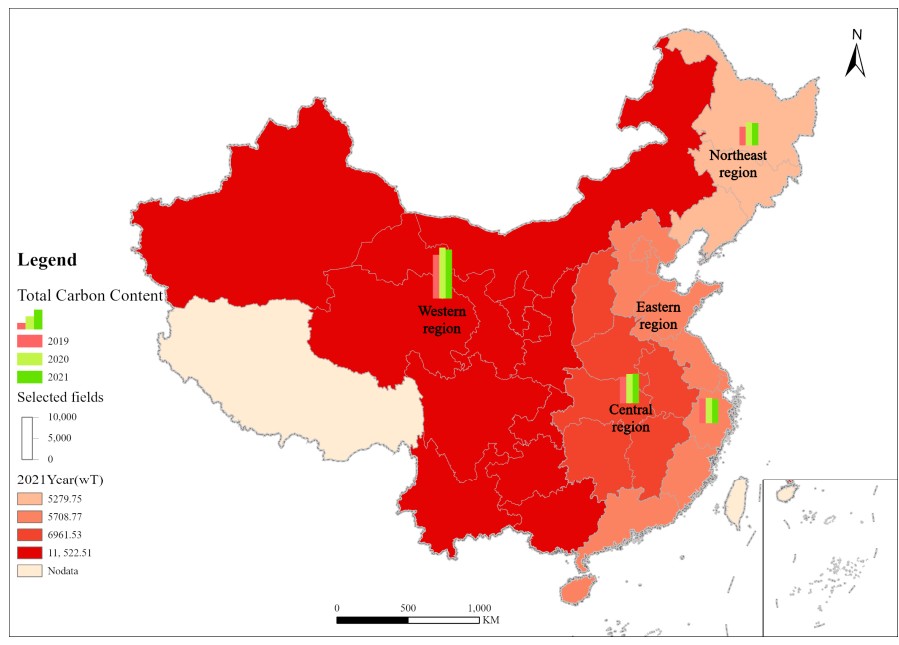

**Figure 2.** Regional distribution of carbon emissions from animal husbandry (2019–2021).

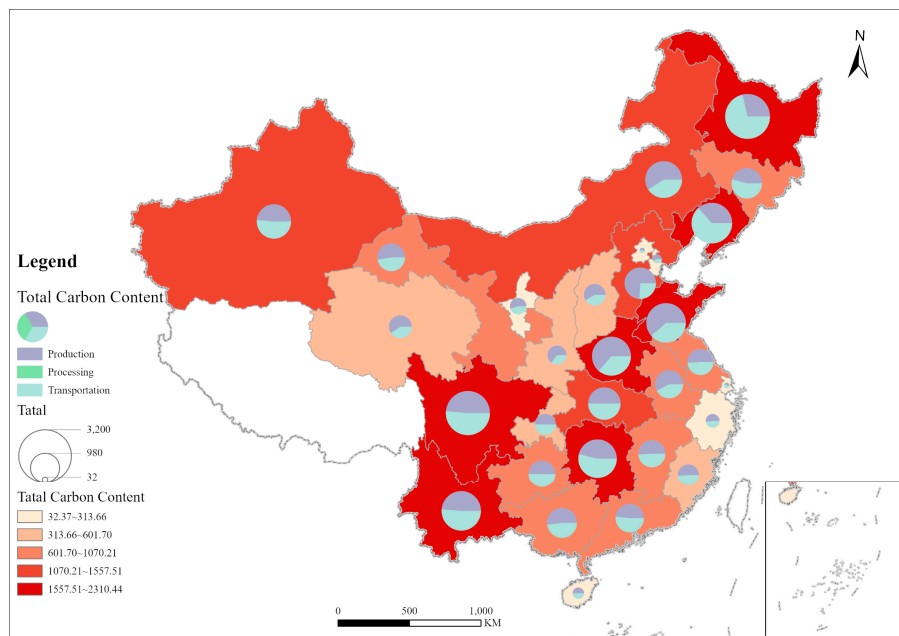

**Figure 3.** Carbon emissions from animal husbandry in China (2021).

### 4.3. Influencing Factors of Carbon Emissions from Animal Husbandry

In Figure 4, the production efficiency of animal husbandry production $(A_1)$, the structural aspect of agriculture $(A_2)$, and the level of urbanization $(A_4)$ are shown to have inhibitory effects on carbon emissions. Particularly, the animal husbandry production efficiency factor $(A_1)$ displays the most prominent inhibitory impact. The main determinants contributing to the increase in carbon emissions from animal husbandry are the per capita agricultural production income factor $(A_3)$ and the total population factor $(A_5)$.

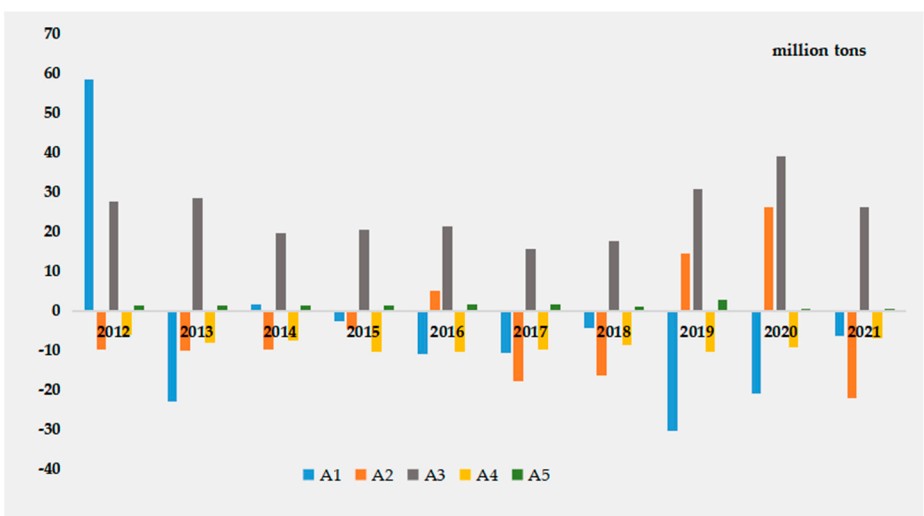

**Figure 4.** Decomposed LMDI factors that affect livestock carbon emissions.

#### 4.3.1. Production Efficiency Factor of Animal Husbandry

The factor of animal husbandry production efficiency $(A_1)$ serves as a factor that reduces carbon emissions from animal husbandry operations. From 2012 to 2021, there was a notable reduction of 109.08 million tons in carbon emissions, largely due to improvements in the efficiency of animal husbandry production processes. The decrease in carbon emissions surpassed the inhibitory impact of agricultural structural adjustments and urbanization. These results suggest that improving the productivity of animal husbandry

production is essential for decreasing carbon emissions from animal husbandry. It also serves as a viable strategy for promoting the low-carbon transition in animal husbandry. Therefore, it is crucial to supervise the shift of animal husbandry from a conventional to an intensive and large-scale production model.

### 4.3.2. Agricultural Structural Factor

The agricultural structural factor $(A_2)$ significantly influences the reduction in carbon emissions within the animal husbandry sector, highlighting its pivotal role in promoting environmental sustainability. Between 2012 and 2021, a decrease of 90.41 million tons in carbon emissions was observed due to agricultural structural adjustments. During the years 2019 and 2020, the proportion of animal husbandry increased continuously in terms of total output value, which comprised agriculture, forestry, animal husbandry, and fishing. During this biennium, agricultural structural elements accounted for 14.32 million tons and 26.11 million tons of carbon emissions in 2019 and 2020, respectively.

### 4.3.3. Per Capita Agricultural Production Income Factor

The per capita agricultural production income factor $(A_3)$ was found to have a positive effect on carbon emissions from animal husbandry, resulting in a total increase of 246.48 million tons from 2012 to 2021. The primary contributor to carbon emissions among the five factors examined is per capita income from agricultural production $(A_3)$. Within the context of a socialist market economy, there has been a consistent rise in the level of awareness of market economy principles among farmers. The economic benefits derived from agricultural production have encouraged farmers to continue engaging in agricultural practices and related downstream industrial activities, thereby resulting in a rise in carbon emissions stemming from animal husbandry.

### 4.3.4. Urbanization Factor

The urbanization factor $(A_4)$ was found to be consistently significant in reducing carbon emissions from animal husbandry, leading to a decrease of 87.01 million tons of carbon emissions over the study period. On the one hand, urbanization has resulted in a constant migration of the rural labor force to urban areas. The per capita area of the rural resident population has experienced a significant increase, thus laying the essential groundwork for the extensive and intensive advancement of animal husbandry. On the other hand, urbanization has also contributed to the progress of economic and social development. The non-agricultural income of farmers has shown an upward trend, whereas the agricultural population involved in animal husbandry has been declining.

### 4.3.5. Total Population Factor

The total population factor $(A_5)$ serves as the primary driver for the increase in carbon emissions from animal husbandry, contributing to a rise of 12.98 million tons of carbon emissions throughout the study period. The large population of China leads to an escalating need for livestock products, thereby leading to an increase in carbon emissions from animal husbandry. Recent shifts in societal attitudes towards fertility, especially among younger individuals, have led to a decrease in fertility rates. This phenomenon has resulted in a diminished influence of overall population growth on the rise of carbon emissions within the sector of animal husbandry.

## 5. Discussion

Carbon emissions from China's livestock industry's production, processing, and transportation stages between 2012 and 2021 were measured in this study using an LCA. The results show that the production stage is the main source of carbon emissions from the livestock industry, including feed cultivation, transportation and processing, livestock feeding, livestock manure management, and gastrointestinal fermentation. Among them, gastrointestinal fermentation and manure management are the main causes of the rise

in carbon emissions during the production stage, accounting for roughly 54.73% of the total emissions during this stage in 2021. Furthermore, carbon emissions from feed grain planting should not be neglected, accounting for about 38.17–41.03% of the total carbon emissions during the livestock production stage from 2012 to 2021. This aligns with the conclusions of several researchers [12,34]. Based on this, the nation should prioritize the application of scientific breeding and the production of low-carbon feed in order to reduce carbon emissions from animal husbandry from the "source". The government should support the growth of large-scale, centralized breeding production; create a large-scale, low-carbon, scientific, and ecologically sustainable circular industrial chain; and implement scientific and technological approaches to lower the carbon emissions of livestock during manure treatment.

Meanwhile, the results of this study show that carbon emissions generated during the transportation stage from 2012 to 2021 accounted for about 45.78–55.21% of the total carbon emissions from the livestock sector. Human consumption activities have made the transportation sector one of the main sources of carbon emissions, and it also shows one of the quickest rates of development, which affects its carbon emissions [35]. Transportation-related carbon emissions are a significant and unavoidable component of the animal husbandry industry [36]. This has a lot to do with the requirement to guarantee the timeliness and freshness of livestock products. In light of this, urban dwellers should be actively encouraged to eat as many local livestock products as possible, along with bolstering advertising and education on low-carbon food consumption, energy savings, and emission reduction. China can simultaneously lower its carbon emissions from the use of livestock products by creating new energy sources, implementing eco-friendly transportation options, and increasing the effectiveness of the existing transportation infrastructure.

The western region of China is consistently ranked highest among the four major regions in terms of total carbon emissions from the livestock industry. This might be because the western area is rich in grassland resources [37]. The growth of animal husbandry is greatly aided by the region's special natural conditions. However, traditional family farming with small-scale operations remains the mainstay for the development and management of animal husbandry [38]. The workforce in the animal husbandry business is primarily composed of middle-aged and older people as a result of the fall in the number of young people in agricultural and herding areas. A low educational attainment, a poor grasp of scientific breeding and environmental conservation, and a general deterioration in the caliber of the livestock work force are the characteristics of this population category. Thus, to support large-scale centralized scientific production and management, the government should enhance the popularization of scientific knowledge about raising livestock and poultry and raise the standard of work performed by practitioners in the livestock industry. Furthermore, local governments should create appropriate policies to draw funds and attract talent, as well as to motivate young people to work in animal husbandry, promote the large-scale and scientific advancement of animal husbandry, and support the restructuring of the industrial system.

From 2012 to 2021, the per capita agricultural production income factor ($A_3$) was found to be the main driver of the increase in carbon emissions from the livestock sector. The advancement of the socialist market economy has led to increasing economic awareness among farmers. This has led to an increase in farmers' motivation to participate in agricultural production, thus becoming an important factor contributing to the increase in carbon emissions from livestock production [39]. In addition, the total population factor ($A_5$) also continues to influence the increase in carbon emissions from animal husbandry. China has a large population base, but the fertility rate has declined in recent years due to a change in the mindset of the younger generation regarding reproduction. As a result, the overall impact of the total population factor on the increase in carbon emissions has weakened. Raising the income level of farmers and herdsmen and narrowing the national income gap is a must for the country's long-term development. However, it must not take the path of development first and environmental management later. It is crucial to focus on

environmental monitoring and management while raising the per capita income of farmers and herders. The awareness of environmental protection among farmers and herdsmen should be raised through education and publicity. At the same time, the efficiency of livestock production should be enhanced through innovative production technology and the promotion of large-scale operations.

Due to the differences in research methods and livestock system boundary delineation, there have been differences in the calculation of carbon emissions from the livestock sector by domestic scholars in recent years [11,12]. Based on a review of previous relevant studies, the calculated results of this study are greater than the results reported by He [26]. This disparity in carbon emissions may be attributed to differences in the size and productivity of the livestock industry. Carbon emissions from China's livestock sector accounted for about 7.4% of China's total carbon emissions in 2021. Due to the availability of data, only a few important livestock products were studied in this study, and carbon emissions from animals such as donkeys, mules, and rabbits were not covered. Consequently, future research should gather more thorough and detailed data by consulting pertinent statistical yearbooks and experiments to supplement the data on carbon emissions from livestock husbandry that were overlooked in this study, and to ensure the accuracy of the estimation results.

## 6. Conclusions

This study utilized LCA to assess carbon emissions from China's livestock industry between 2012 and 2021. Additionally, the LMDI decomposition method was employed to analyze the influence of five factors on carbon emissions from China's livestock industry. This approach aimed to examine the individual contribution of each factor to the increase in carbon emissions. The research results can be summarized as follows:

(1) From 2012 to 2021, there was an increase in total carbon emissions from China's livestock industry from 287.74 million tons to 294.73 million tons, representing an average annual growth rate of 0.42%. According to the total carbon emissions of each stage in 2021, the order from largest to smallest is as follows: the production stage (149.61 million tons), the transportation stage (145.07 million tons), and the processing stage (0.049 million tons).

(2) The ranking of average annual total carbon emissions by region is as follows: western region (109.58 million tons) > central region (69.81 million tons) > eastern region (60.94 million tons) > northeast region (42.01 million tons). The western region is predominantly characterized by extensive animal husbandry as it benefits from abundant grassland resources.

(3) From 2012 to 2021, the largest contributing factor to the increase in livestock carbon emissions was per capita income from agricultural production $(A_3)$. Furthermore, the total population factor $(A_5)$ remained a significant driver of the increase in livestock carbon emissions.

**Author Contributions:** Methodology, C.P.; investigation, X.W.; data curation, X.W.; writing—original draft preparation, X.W.; writing—review and editing, X.X. and Y.W.; funding acquisition, C.P. All authors have read and agreed to the published version of the manuscript.

**Funding:** This research was funded by the National Social Science Fund Training Program of Southwest University of Science and Technology, the Fundamental Research Funds for the Central Universities, and the Natural Science Foundation of Sichuan Province. The grant numbers are 23sxb073, XJ2023004401, and 24NSFSC7124, respectively.

**Institutional Review Board Statement:** Not applicable.

**Informed Consent Statement:** Not applicable.

**Data Availability Statement:** Data are not available due to privacy or ethical restrictions.

**Conflicts of Interest:** The authors declare no conflicts of interest.

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
