# Peer review of "Assessing Carbon Emissions from Animal Husbandry in China: Trends, Regional Variations and Mitigation Strategies"

_sustainability, doi:10.3390/su16062283_

Round 1

Reviewer 1 Report

Comments and Suggestions for Authors

The presented research is characterized by a high degree of relevance, is applied in nature and corresponds in its subject matter to the scope of the journal. The article has a clear logical structure, the sections of the study meet the requirements for presenting the results of scientific research. The selected methods and a comprehensive study of emissions from livestock by stages of the life cycle and the factors determining it are quite informative and deserve support. The empirical calculations correspond to the theoretical concept. At the same time, there are a number of points of discussion and comments, the elimination of which will improve the work. From a substantive point of view, there are several questions for the authors. First, a paper that is being considered in 2024 indicates that: By 2021, the output value of 40 of animal husbandry in China is projected to reach 3,991.08 billion yuan, representing

27.15% of the total agricultural output value in the country. In this case, it probably makes sense not to talk about the forecast, as the specified period has already come. also in the abstract and the text of the work itself, the authors write about the integration of the three industries in China, but do not specify which industries they are talking about and how this affects the problem considered in the work. In some cases, the wording in the text raises a question. We are talking about moving towards a green economy, while using phrases regarding carbon emissions: In 2020, the nation

proposed the dual-carbon objective of achieving "carbon neutrality by 2030 and reaching peak emissions by 2060", This analysis aims to provide a scientific basis for China's animal husbandry to achieve a green and low-carbon transformation at the industry level, and to contribute to the realization of peak CO2 emissions and carbon neutrality. Should we really be talking about a peak or a minimum? It seems advisable to remove the letter designations of the factors in the annotation. Due to technical errors, a significant part of the sources are not displayed correctly, which complicates the assessment of their compliance with the considered aspects of the problem. The links to empirical data sources themselves appear in the text after the data and calculations themselves, which requires correction. The sequence of table references is not entirely clear (first 1, then 3, then 2). Table 1 shows the cattle and cow positions separately. Whereas, according to standard classifications, cows belong to cattle. It also raises the question of the choice of factors affecting emissions. In this case, a more detailed argumentation is required. For example, why, if technological advancements have a positive impact, are they not taken into account by the authors? The question also arises, why is the impact factor analysis not carried out taking into account the stages of the life cycle? Taking into account the stated goals, it seems advisable to formulate more detailed policy and management recommendations in terms of discussion. The originality of the work (according to the anti-plagiarism system) is 66.69%, and citations are 15.34%, it is recommended to improve these indicators. In general, the answer to these questions will further improve the quality of the presented scientific work.

Comments on the Quality of English Language

Some formulations raise questions, perhaps due to linguistic difficulties. For example, peak or minimum

Author Response

请参阅附件。

Reviewer 2 Report

Comments and Suggestions for Authors

This article presents a scientific approach to solving the social problem of CO2 emissions, which determines dynamic climate changes. The article presents recommendations for the animal breeding process that support its transformation in order to reduce CO2 emissions and achieve neutrality. An important contribution of this article is to take into account the complex nature of the problem of CO2 emissions (emissions related to production, processing and transport) and to provide a thorough presentation of the theoretical background for the estimates made. The strength of this study is the identification of what factors and strategies can contribute to achieving CO2 neutrality.

The scientific problem is clearly defined, reasonably well described, and is relevant to the field. The knowledge gap has been precisely identified. However, in this form the manuscript is not transparent and requires corrections and improvement of its organization.

The weak point of the article in this form is the inability to check the reliability of data sources for calculations. The article requires editorial correction. Editing improvement of the article (source items should be assigned to appropriate fragments of the article content) will allow you to assess the reliability of the estimates obtained and, therefore, the recommendations formulated on their basis. The data sources for estimates made based on well-described methods should be indicated more precisely. In the version of the article being assessed, only a few fragments have easily identifiable references to the literature. In many places you can only see the sentence: "Error! Reference source not found.” This needs improvement.

Table 1 could be improved to make it easier to interpret and understand. Please explain why Table 1 includes data for both „cattle” and „cow”. Is there a justification for a special approach to "cow", which in my opinion is a component of "cattle"?

I believe that the scientific basis of the manuscript could be improved by also providing definitions in section 4.3. In my opinion, synthetic definitions should be provided: "Animal husbandry production efficiency factor", "Agricultural structural factor", "Per capita agricultural production income factor", "Urbanization factors", "Total population factor". These concepts (mainly A2) are not commonly used, so a brief description of them will facilitate the perception of the message.

In point 4.3.4, the sentence (lines 384-386) is incomprehensible to me: „The per capita area of rural resident population has significantly expanded, thereby establishing the fundamental prerequisites for the extensive and intensive development of animal husbandry.”

Some conclusions are not consistent with the evidence and arguments presented. There is a contradiction between what is described in section 4.3.4 (lines 381-383) indicating that: „The urbanization factor (A4) has consistently played a significant role in mitigating the carbon emissions from animal husbandry, resulting in a reduction of 87.01 million tons of carbon emissions during the study period.”, and what is described next (lines 436-438) indicating that: „Furthermore, the urbanization factor (A4) and the total population factor (A5) persist in influencing the increase in carbon emissions from animal husbandry.”.

Almost half (14/33) of the cited sources are recent publications (from the last 5 years). I estimated the number of self-citations at approximately 12%. I think the number of self-citations can be reduced.

Despite the comments submitted, I emphasize that the problem presented in this article is still current and interesting for the scientific community. All the more so because the recommendations resulting from this study are partly in contradiction with the recommended and ongoing changes in animal breeding in European countries. What I mean here are new trends manifesting themselves in moving away from intensive agriculture towards local agriculture and attaching importance to ensuring animal welfare. Therefore, I think it would be worth discussing this topic in this article. Such a discussion will make it more valuable and mature.

In addition, it is necessary to discuss the problem under study based on your own results and those of other researchers (references to source materials should be provided), as well as trends and changes taking place in various regions of the world (references to source materials should be provided). At the moment, the "Discussion" chapter basically presents only conclusions. These should constitute a separate chapter called "Conclusions".

Reviewer 3 Report

Comments and Suggestions for Authors Thank you for the opportunity to review the article. The article discusses a very important and current topic of greenhouse gas reduction, which, as the authors themselves emphasize, has become an urgent global problem for all countries. In addition, the agricultural sector, which is a source of greenhouse gas emissions such as methane, nitrous oxide and carbon dioxide, is important from the point of view of the topic discussed. However, the article has several areas that should be completed before its publication. Suggestions are provided below:   1. Introduction: In this part, there is a need to clarify the research gap, supported by the literature. Moreover, it is not usually practiced to include information related to the description of the research method in the introduction - this type of information should be included in the materials and methods section. Additionally, instead of the goal, hypotheses or research problems should be indicated,   2. Literature review: In this part, a lot of attention was devoted to justifying the validity of the use of the research method and the factors taken into account. However, there is no indication: Have other scientists already conducted research on the volume/reduction of greenhouse gases in the study area? What were the main conclusions from this research? It is worth adding such information to this part. Additionally, the text of this section includes the messages "Error! Reference source not found” which should be corrected,   3. Materials and methods: This part is well prepared, but you should review the explanations of the presented formulas again, because not all of them are clear, e.g. in formula 3 it is not explained what GWPch4 is, while in formula 7 it is not known whether it is EFC. Additionally, "Error!" messages appear in the text and tables. Reference source not found” which should be corrected,   4. Results: This part is well prepared, however in Figure 1 there are 3 colors in the legend and the bars in the chart only have 2 colors - this should be explained in the description.   5. Discussion: In this part, the obtained results should be compared to the research of other authors (with references to the literature). Moreover, this section should also present the limitations of the study conducted as well as directions for future research.            

Reviewer 4 Report

Comments and Suggestions for Authors

SPECIFIC COMMENTS:

Title

The title of the paper includes the concept of Mitigation Strategies; However, nowhere in the document are arguments presented about this concept (see comment below in the Discussion section).

Abstract

Lines 12-13: I ask the authors to clarify which are the three mentioned industries in China.

Keywords: I suggest that the authors eliminate the keywords animal husbandry and carbon emissions, which are in the title. Instead, I suggest the keywords GHG emissions, LMDI method, LCA method, driving factors carbon emissions, livestock production stages.

2 Literature review

No comment

3. Material and Methods

Tabla 1: Cow is Dairy cattle?

The title of Table 2 is incomplete, it should be CH4 emission and include fecal management (for example: “CH4 emission coefficient of gastrointestinal and fecal management”

In general, the numbering of sections and subsections of the document is very confusing and is also not uniform, which makes reading the manuscript difficult. For example, in the Materials and Methods section the authors use parentheses and half-parentheses to separate subsections, for example (1) or 1) and in the Results section they use numerals without parentheses. Therefore I suggest that authors number all sections and subsections of the manuscript using only numerals (e.g. 1.; 1.1.; 1.1.1. etc.).

In section (2) Livestock and poultry raising, I request that the authors include background additional information on the predominant characteristics of animal production systems practiced in China. Are they industrial, technical or intensive systems? Are they family systems? Are they extensive systems?

In section 3.3, authors should include the species and total number of animals, at least in the most recent year of the analysis (2022) or in the period 2012-2021.

On the other hand, the separation of some sections and subsections in the manuscript makes it difficult to read. Therefore, I suggest:

Separate line 167 from the preceding paragraph

Separate line 190 from the preceding Table 1

Separate line 203 from the preceding paragraph

Separate line 221 from the preceding Table 2

Separate line 233 from the preceding paragraph

Separate line 277 from the preceding paragraph

Separate line 248 from the preceding Table 3

Lines 288-289: The words animal husbandry are repeated

It says: “The data for this study, including the output value of animal husbandry, agriculture, forestry, animal husbandry, and fishery,…”

It should say: “The data for this study, including the output value of animal husbandry, agriculture, forestry, and fishery,…”

4. Resultados

There are some contradictions in the statements presented in the Results and Discussion sections (see comments in the following Discussion section).

5. Discusión

The urbanization factor (A4) is not influencing the increase in carbon emissions from animal husbandry; therefore, I ask the authors to review and correct the following statements:

Lines 345-348 (Results): “the animal husbandry production efficiency factor (A1), agricultural structural factor (A2), and urbanization factor (A4) exhibit significant inhibitory impacts on carbon emissions,…”

Lines 381-382 (Results): “The urbanization factor (A4) has consistently played a significant role in mitigating the carbon emissions from animal husbandry,…”

Lines 436-438 (Discussion): “Furthermore, the urbanization factor (A4) and the total population factor (A5) persist in influencing the increase in carbon emissions from animal husbandry.”

On the other hand, the discussion focuses exclusively on the situation in China, but I consider that the article would be complemented if, in addition to discussing their results from the perspective of their country, the authors discuss their obtained results with the carbon emissions from animal husbandry in other countries.

In this regard, I request that the authors answer the following questions: How much CO2 does livestock husbandry in China emit compared to other countries?

How important are CO2 emissions from Chinese livestock husbandry to total global emissions?

Although the concept of Mitigation Strategies is included in the title of the paper, the authors do not include comments on this concept throughout the document. Therefore, I request that the authors discuss specific mitigation strategies related to the results obtained in the research, mainly mitigation strategies in the production stage (enteric CH4 emissions from livestock, carbon emissions from manure management system and energy consumption in livestock and poultry raising), as well as mitigation strategies in the transportation stage.

Additionally, in the Discussion section I suggest that the authors include the limitations of the study.

I also suggest that authors include a Conclusions section in the manuscript.

References

The majority of references are not adequately cited in the text, so it is not possible to verify their inclusion in the reference list.

Starting from Reference 9 the legend appears: “Error! Reference source not found” Therefore, authors should review each of the citations in the text and verify them with the reference list.

Reviewer 5 Report

Comments and Suggestions for Authors

The authors attempt to investigate the increased greenhouse gas (GHG) emissions from animal husbandry, exacerbating climate warming and imposing pressure on global environmental protection.

Although interesting maybe, I have the following comments:

1.          Given the scope of the journal the authors should place the whole discussion under the notion of sustainable development.   Thus, a short discussion of the term should be provided in the introduction.   In this vein, the following two papers should be included.   (a) “Broad strokes towards a grand theory in the analysis of sustainable development: a return to the classical political economy”, New Political Economy, 27(5), pp. 866-878, and (b) “The concept of sustainable development: From its beginning to the contemporary issues”, Zagreb International Review of Economics & Business, 21(1), 67-94.

2.          No in-text citations included in the text!

3.          The discussion section should be expanded included analytical comparisons with the existed bibliography.

4.          Methodology section should be included.

5.          Conclusions should, also, be included, with possible limitations of their study.

Comments on the Quality of English Language

 Extensive editing of English language required

Round 2

Reviewer 1 Report

Comments and Suggestions for Authors

Thank you for the detailed answers and the adjustments made

Author Response

 Thank you for your invaluable comments that made the revised manuscript more readable and scientifically sound!

Reviewer 2 Report

Comments and Suggestions for Authors

Line 271 – GWPCH4 (the CH4 symbol should be written in capital letters)

Line 307 – TCP – no explanation of this symbol in the text

Line 329 – TCS symbol was placed instead of TCT symbol

Reviewer 3 Report

Comments and Suggestions for Authors

Thank you for making efforts to improve the article and taking into account the proposed suggestions. However, there are still a few aspects worth paying attention to before publishing an article:

1. The need for research is well presented in section 2.3, but there are no references to the literature. The authors wrote, for example, "Nevertheless, the majority of recent research on estimating carbon emissions from animal husbandry concentrates on the breeding stage of livestock and poultry, paying little attention to downstream processing, particularly transportation." - it is worth providing sources for the above-mentioned sentences latest research.

2. In their responses to the reviewer, the authors wrote that in point 2.3. research problems and hypotheses were addressed. After reviewing this section, no research problems or hypotheses were found. It's worth supplementing it.

3. In the discussion part (lines 496, 503, 549) there are the sentences "This aligns with the conclusions of several researchers34"; “According to some academics…”;” Due to differences in research methods and livestock system boundary delineation,… “ – in such cases, more than one source of literature should be provided. Moreover, in the sentence of line 552, I think something is missing, it says "…those of He"

4. The way of writing sources should be improved throughout the text. According to the magazine's requirements, numbers should be given in brackets.

Reviewer 4 Report

Comments and Suggestions for Authors

The authors must review and correctly write citations in the text

I request that the authors include specific mitigation strategies related to the results obtained, mainly mitigation strategies in the production stage (enteric CH4 emissions from livestock, carbon emissions from manure management system and energy consumption in livestock and poultry raising), as well as mitigation strategies in the transportation stage.

Reviewer 5 Report

Comments and Suggestions for Authors

the manuscript is now publishable.

Comments on the Quality of English Language

only minor
